# STRUCTURED INITIALIZATION FOR ATTENTION IN VISION TRANSFORMERS

## ABSTRACT

The application of Vision Transformers (ViTs) to new domains where an inductive bias is known but only small datasets are available to train upon is a growing area of interest. However, training ViT networks on small-scale datasets poses a significant challenge. In contrast, Convolutional Neural Networks (CNNs) have an architectural inductive bias enabling them to perform well on such problems. In this paper, we propose that the architectural bias inherent to CNNs can be reinterpreted as an initialization bias within ViT. Specifically, based on our theoretical findings that the convolutional structures of CNNs allow random impulse filters to achieve performance comparable to their learned counterparts, we design a "structured initialization" for ViT with optimization. Unlike conventional initialization methods for ViTs, which typically (1) rely on empirical results such as attention weights in pretrained models, (2) focus on the distribution of the attention weights, resulting in unstructured attention maps, our approach is grounded in a solid theoretical analysis, and builds structured attention maps. This key difference in the attention map empowers ViTs to perform equally well on small-scale problems while preserving their structural flexibility for large-scale applications. We show that our method achieves significant performance improvements over conventional ViT initialization methods across numerous small-scale benchmarks including CIFAR-10, CIFAR-100, and SVHN, while maintaining on-par if not better performance on large-scale datasets such as ImageNet-1K.

## 1 INTRODUCTION

Vision Transformers (ViTs) have shown remarkable performance with large-scale training data. However, their performance significantly drops when applied to small-scale datasets. To close this performance gap, various methods have been proposed, including self-supervised pre-training in large-scale datasets (Dosovitskiy et al., 2021; Touvron et al., 2021), advanced data augmentation techniques (Yun et al., 2019; Cubuk et al., 2020), and incorporation of convolutional layers (Wu et al., 2021; Liu et al., 2021; Yuan et al., 2021; Li et al., 2023; Dai et al., 2021). Recently, Zhang et al. (2022) explored leveraging pretrained weights to initialize ViTs. Building on this pioneering idea, researchers found that a simple network weight initialization can improve ViT training on small-scale datasets without altering ViT architectures. In particular, Trockman & Kolter (2023) developed a "mimetic" method that mimics the distribution of pretrained ViT weights during initialization. Nevertheless, this distribution relies on the observations from pretrained models with the same architectures. Similarly, Xu et al. (2024) proposed to directly sample weights from pretrained large-scale models, while it is not practical to pretrain a large model before deploying a small-scale one. In conclusion, these initialization strategies (1) focus more on replicating the distribution of attention weights rather than the structure of attention maps, and (2) rely on pretraining with large-scale datasets, which are often not readily available for many domain-specific applications.

In this work, we introduce *structured initialization* for ViTs, which focuses on structuring attention maps and does not rely on any form of pretraining with large-scale models. This initialization strategy is grounded in our theoretical findings that the structural bias within randomly initialized depth-wise (spatial mixing) convolutions is the key factor enabling comparable performance (Cazenavette et al., 2023) to their learned counterparts in ConvMixer (Trockman & Kolter, 2022) and ResNet (He et al., 2016) frameworks. We further propose to use random impulse convolution kernels as the initialization structure for attention maps, as shown in Fig. 1. In detail, we use a simple yet efficient optimization

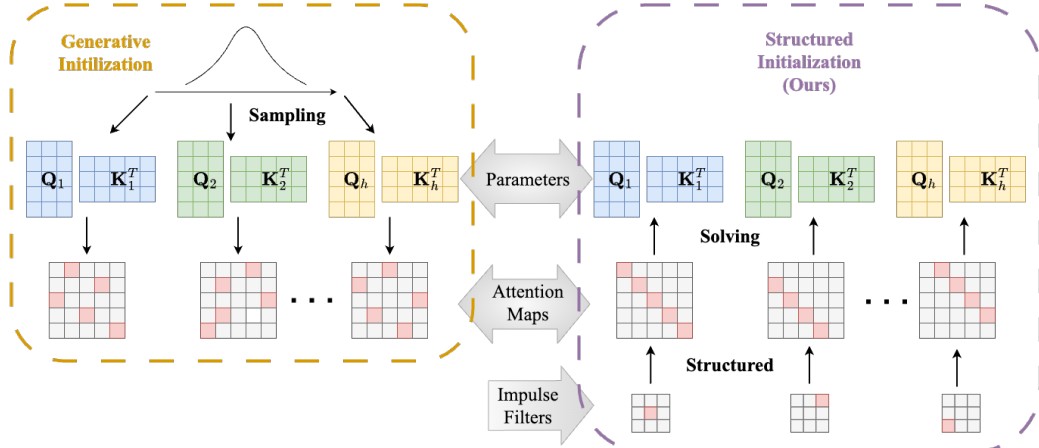

Figure 1: Illustration of conventional generative initialization and structured initialization (ours) strategies for the weights $\mathbf{Q}$ and $\mathbf{K}$ of the attention map in transformers. Conventional generative initialization involves sampling parameters $\mathbf{Q}$ and $\mathbf{K}$ from certain distributions, such as Gaussian or Uniform, resulting in unstructured initial attention maps. In contrast, our structured initialization strategy imposes constraints on the structure of the initial attention maps, specifically requiring them to be random impulse filters. The initialization of parameters $\mathbf{Q}$ and $\mathbf{K}$ is computed based on this requirement on attention maps. Note that in both attention maps and random impulse filters, the pink cells indicate ones, while the gray cells represent zeros.

to solve for attention weights such that the resulting attention maps are structured as convolution filters. Benefiting from locality assumptions in CNNs, our method enables effective training for ViTs on small-scale datasets. Additionally, different from methods that directly introduce convolutions into attention (Yuan et al., 2021; Li et al., 2023; Dai et al., 2021), our method does not change the architecture of ViTs, maintains the inherent structural flexibility in ViTs.

To conclude, our paper makes the following contributions:

– We build a conceptual link between the structural bias in CNNs and initialization in ViTs, and provide a solid theoretical analysis for the effectiveness of using random convolution filters as initialized attention maps.

– To the best of our knowledge, we are the first to focus on the structure of attention maps for ViT initialization and propose embedding the structural inductive bias of CNNs as an initialization bias within ViTs without changing the Transformer architectures.

– We further demonstrate state-of-the-art performance for ViT training across various small-scale datasets including CIFAR-10, CIFAR-100, and SVHN, while achieving competitive performance on large-scale datasets such as ImageNet-1K.

## 2 RELATED WORK

**Introducing inductive bias of CNN to ViT through architecture.** Many efforts have aimed to incorporate convolutional inductive bias into ViTs through architectural modifications. Dai et al. (2021) proposed to combine convolution and self-attention by mixing the convolutional self-attention layers. Pan et al. (2021) and Li et al. (2023) introduced hybrid models wherein the output of each layer is a summation of convolution and self-attention. Wu et al. (2021) explored using convolution for token projections within self-attention, while Yuan et al. (2021) demonstrated promising results by inserting a depthwise convolution before the self-attention map as an alternate strategy for injecting inductive bias. d'Ascoli et al. (2021) introduced gated positional self-attention (GPSA) to imply a soft convolution inductive bias. All these efforts have been proven to be effective. However, these techniques have a fundamental limitation—they aim to introduce the inductive bias of convolution through architectural choices. Our approach, on the other hand, stands out by not requiring any modifications of architecture. Such an approach offers several advantages as it: (i) exhibits data

efficiency on small-scale datasets, (ii) retains the architectural freedom to be seamlessly applied to larger-scale datasets, and (iii) gives an alternate theoretical perspective on how the inductive bias of convolution can be applied within transformers.

**Initializing ViT from pretrained weights.** To date, the exploration of applying inductive bias through initialization within a transformer is limited. Zhang et al. (2022) posited that the benefit of pretrained models in ViTs can be interpreted as a more effective strategy for initialization. Trockman & Kolter (2023); Trockman et al. (2022) recently investigated the empirical distributions of self-attention weights, learned from large-scale datasets, and proposed a mimetic initialization strategy. While this approach lies between structured and generative initialization, it relies on the pretraining results of large models. Xu et al. (2024) directly sampled weights from pretrained large-scale models as initialization for smaller models. This may be astonishing at first glance, but the sampled weights must follow the distribution of these pretrained weights, which means this method is a special case of mimetic initialization. A key difference in our approach is that our method does not require offline knowledge of pretrained networks (mimetic or empirical). Instead, our initialized structure is derived from theoretical analysis of convolution layers.

**Convolution as attention.** Since their introduction (Vaswani et al., 2017; Dosovitskiy et al., 2021), the relationship between transformers and CNNs has been a topic of immense interest to the vision community. Andreoli (2019) studied the structural similarities between attention and convolution, bridging them into a unified framework. Building on this, Cordonnier et al. (2020) demonstrated that self-attention layers can express any convolutional layers through a careful theoretical construction. While these studies highlighted the functional equivalence between self-attention in ViTs and convolutional spatial mixing in CNNs, they did not delve into how the inductive bias of ViTs could be adapted through this theoretical connection. In contrast, our work offers a simple insight: random convolutional impulse filter can be effectively approximated by softmax self-attention.

## 3 WHY RANDOM FILTERS WORK?

Cazenavette et al. (2023) recently demonstrated remarkable performance of randomly initialized convolution filters in ConvMixer and ResNet when solely learning the channel mixing parameters. However, they failed to offer any insights into the underlying reasons. In this section, we provide a theoretical analysis of how solely learning channel mixing can be sufficient for achieving reasonably good performance. Our theoretical findings are significant as they establish a conceptual link between the architecture of ConvMixer and the initialization of ViT, offering a deeper understanding of desired properties for spatial mixing matrices. Without losing generality, we have omitted activations (*e.g.*, GeLU, ReLU, *etc.*), bias, batch normalization, and skip connections in our equations for clarity.

**Remark 1** *Let us define the patch embeddings or intermediate layer outputs in ConvMixer as* $\mathbf{X} = [\mathbf{x}_1, \mathbf{x}_2, \ldots, \mathbf{x}_D]$, *where* $D$ *is the number of channels and* $N$ *is the number of pixels in the vectorized patch* $\mathbf{x} \in \mathbb{R}^N$. *An interesting observation is the rank (stable rank, defined as* $\sum \sigma^2 / \sigma_{max}^2$) *of* $\mathbf{X}$ *is consistently much smaller than the minimum dimension* $\min(N, D)$ *of* $\mathbf{X}$, *indicating a significant amount of redundancy in patch embeddings or intermediate layer outputs in deep networks.*

Let us define a 2D convolution filter as $\mathbf{h} \in \mathbb{R}^{f \times f}$. In general, this kernel can be represented as a circulant matrix $\mathbf{H} \in \mathbb{R}^{N \times N}$, such that $\mathbf{h} * \mathbf{x} = \mathbf{H}\mathbf{x}$, where $*$ is the convolutional operator. The relation between the convolutional matrix and convolution filters is explained in detail in Appendix B. A ConvMixer block $\mathbf{T}^{\text{Conv}} : \mathbb{R}^{N \times D} \to \mathbb{R}^{N \times D}$ is composed of a spatial mixing layer $\mathbf{T}_S^{\text{Conv}} : \mathbb{R}^{N \times D} \to \mathbb{R}^{N \times D}$ and a channel mixing layer $\mathbf{T}_C^{\text{Conv}} : \mathbb{R}^{N \times D} \to \mathbb{R}^{N \times D}$, where $\mathbf{T}_S^{\text{Conv}}$ is defined by a sequence of convolution filters $\mathbf{H} = [\mathbf{H}_1, \mathbf{H}_2, \ldots, \mathbf{H}_D] \in \mathbb{R}^{D \times N \times N}$, $\mathbf{H}_i \in \mathbb{R}^{N \times N}$, and $\mathbf{T}_C^{\text{Conv}}$ is defined by a weight matrix $\mathbf{W} \in \mathbb{R}^{D \times D}$. With input $\mathbf{X} = [\mathbf{x}_1, \mathbf{x}_2, \ldots, \mathbf{x}_D] \in \mathbb{R}^{N \times D}$, one ConvMixer block can be represented as

$$\mathbf{T}_S^{\text{Conv}}(\mathbf{X}; \mathbf{H}) = [\mathbf{H}_1\mathbf{x}_1, \mathbf{H}_2\mathbf{x}_2, \ldots, \mathbf{H}_D\mathbf{x}_D], \tag{1}$$

$$\mathbf{T}_C^{\text{Conv}}(\mathbf{X}; \mathbf{W}) = \mathbf{X}\mathbf{W}, \tag{2}$$

$$\mathbf{T}^{\text{Conv}}(\mathbf{X}) = \mathbf{T}_C^{\text{Conv}}(\mathbf{T}_S^{\text{Conv}}(\mathbf{X}; \mathbf{H}); \mathbf{W}) = [\mathbf{H}_1\mathbf{x}_1, \mathbf{H}_2\mathbf{x}_2, \ldots, \mathbf{H}_D\mathbf{x}_D]\mathbf{W}. \tag{3}$$

**Definition 1** *For a set of vectors* $\mathcal{V} = \{\mathbf{v}_i\}_{i=1}^N$, *we say that* $\mathcal{V}$ *is* $M - k$ *spanned if there exists a way to split* $\mathcal{V}$ *into* $k$ *non-overlapping subsets, such that each subset spans the same* $M$-*dimensional space.*

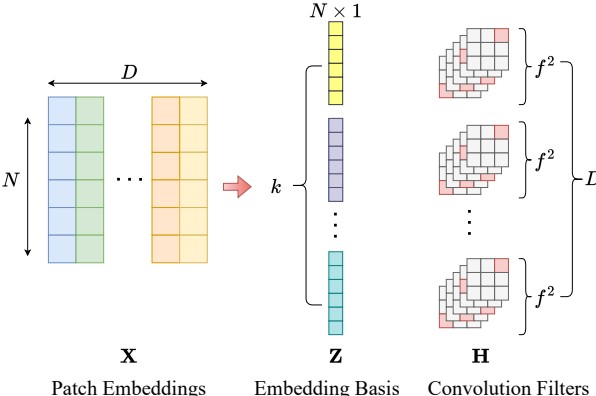

Figure 2: Illustration of why random spatial convolution filters are effective. Patch embeddings $\mathbf{X} \in \mathbb{R}^{N \times D}$ are typically rank-deficient and can be approximately decomposed to $k$ basis. Meanwhile, a linear combination of $f^2$ linearly independent filters $\mathbf{h}$ can express any arbitrary filter in the filter space $\mathbb{R}^{f \times f}$. Based on these two observations, we derive the inequality $D \geq kf^2$ in Proposition 1.

**Proposition 1** *A ConvMixer block $\mathbf{T}$ consists of a spatial mixing layer $\mathbf{T}_S$ with convolution filters $\mathbf{H}$ and a channel mixing layer $\mathbf{T}_C$. Let $D$ be the number of channels, $k$ be the rank of input $\mathbf{X}$, and $\mathbf{H}$ be $M - k$ spanned. For any $\mathbf{T}'$ composed of $\mathbf{T}'_C$ and $\mathbf{T}'_S$, where $\mathbf{T}'_S$ is defined by a $\mathbf{H}'$ in the $M$-dimensional space spanned by $\mathbf{H}$, there always exists a channel mixing layer $\mathbf{T}_C$ with weight $\mathbf{W}$ such that $\mathbf{T}(\mathbf{X}) = \mathbf{T}'(\mathbf{X})$.*

For simplicity, we include the full proof in Appendix A. Note that since $H$ are convolution matrices, their span lies in $\mathbb{R}^{f \times f}$ instead of $\mathbb{R}^{N \times N}$, where $f$ is the kernel size. In practice, $D \geq kf^2$, indicating that randomly initialized spatial convolution kernels are $f^2 - k$ spanned and satisfy Proposition 1. Consequently, the trained results of $\mathbf{T}'_C$ and $\mathbf{T}'_S$ can be achieved by solely training the channel mixing weights $\mathbf{W}$ of $\mathbf{T}_C$ in $\mathbf{T}$, while keeping a fixed spatial mixing layer $\mathbf{T}_S$. Hence the following corollaries can be obtained, and Corollary 1 explains the phenomenon Cazenavette et al. (2023) found, mentioned at the beginning of the section. The related experimental evidence is given in Sec. 5.5.

**Corollary 1** *Random initialized spatial convolution filters perform as well as trained spatial convolution filters since the $f^2 - k$ spanned condition in Proposition 1 is satisfied.*

**Corollary 2** *Random impulse spatial convolution filters perform as well as trained spatial convolution filters since the $f^2 - k$ spanned condition in Proposition 1 is satisfied.*

**Corollary 3** *Spatial convolution filters with all ones (referred to as "box" filters) can only produce averaging values since they are $1 - k$ spanned.*

# 4 STRUCTURED INITIALIZATION FOR ATTENTION MAP

## 4.1 EXPECTED INITIALIZED ATTENTION MAP STRUCTURE

ConvMixer and ViT share most components in their architectures. The gap in their performance on small-scale datasets stems from their architectural choices regarding spatial mixing matrix. Although depthwise convolution (ConvMixer) and multi-head self-attention (ViT) may appear distinct at first glance, their underlying goal remains the same: to identify spatial patterns indicated by the spatial mixing matrix. As defined in Sec. 3, similar to the spatial mixing step in ConvMixer defined in Eq. (1), the spatial mixing step of multi-head attention can be expressed as

$$\mathbf{T}_S^{\text{ViT}}(\mathbf{X}; \mathbf{M}) = [\mathbf{M}_1\mathbf{x}_1, \ldots, \mathbf{M}_1\mathbf{x}_d, \mathbf{M}_2\mathbf{x}_{d+1}, \ldots, \mathbf{M}_2\mathbf{x}_{2d}, \ldots, \mathbf{M}_h\mathbf{x}_{(h-1)d+1}, \ldots, \mathbf{M}_h\mathbf{x}_{h*d}], \quad (4)$$

where $d$ represents the feature dimension in each head, typically set to $D/h$, with $h$ being the number of heads, and the matrices $\mathbf{M}_i$ for multi-head self-attention can be expressed as follows:

$$\mathbf{M}_i = \text{softmax}(\mathbf{X}\mathbf{Q}_i \mathbf{K}_i^T \mathbf{X}^T), \quad (5)$$

where $\mathbf{Q}_i$, $\mathbf{K}_i \in \mathbb{R}^{D \times d}$ denote the attention weight matrices.

It is worth noting that in Eq. (1), the spatial matrices $\mathbf{H}$ are in convolutional structure, resulting in a span of $\mathbb{R}^{f \times f}$ instead of $\mathbb{R}^{N \times N}$, despite each $\mathbf{H}_i$ having a size of $N \times N$. This structural constraint ensures that CNNs focus on local features but struggle to capture long-range dependencies. In contrast, the span of spatial matrices $\mathbf{M}$ in Eq. (4) is $\mathbb{R}^{N \times N}$, allowing for greater learning capacity without these limitations. However, a random initialized $\mathbf{Q}$ and $\mathbf{K}$ contain no structural information, resulting in random matrices as depicted in the bottom left of Fig. 1.

Leveraging this insight, we propose to initialize the attention map for each head in ViT to a convolutional structure as denoted in the bottom right of Fig. 1. Our initialization strategy preserves both the advantage of locality and the capacity to learn long-range information. For clarity and brevity, the following discussions will focus only on one head of multi-head self-attention. Therefore, from Eq. (4) and Eq. (1), our structured initialization strategy can be represented as

$$\mathbf{T}_S^{\text{ViT}}(\mathbf{X}; \mathbf{M}) \approx \mathbf{T}_S^{\text{Conv}}(\mathbf{X}; \mathbf{M}) \Rightarrow \mathbf{M}_{\text{init}} = \text{softmax}(\mathbf{X}\mathbf{Q}_{\text{init}}\mathbf{K}_{\text{init}}^T\mathbf{X}^T) \approx \mathbf{H}. \tag{6}$$

**Why using impulse filters?** Usually, random convolution filters contain both positive and negative values, while the output of the softmax function is always positive. One straightforward option is to use random positive convolution filters with a normalized sum of one, following the property of softmax. However, this approach often proves inefficient as the patterns may be too complicated for a softmax function to handle. Tarzanagh et al. (2023) found that the softmax attention map functions as a feature selection mechanism, and typically tends to select a single related feature. In convolution filters, this softmax attention map can be viewed as an impulse filter. According to Proposition 1, random impulse filters are also $f^2 - k$ spanned. In conclusion, when initializing a softmax attention map, the most straightforward and suitable choice is random impulse convolution filters.

**Pseudo input.** The advantage of self-attention is that its spatial mixing map is learned from data. The real input to an attention layer is $\mathbf{P} + \mathbf{X}$ for the first layer and $\mathbf{X}$ (the intermidiate output from previous layer) for the following layers. However, during initialization, there is no prior information about the input. To address this problem, we simply treat all the layers identically and use absolute sinusoidal positional encoding $\mathbf{P}$ (Dosovitskiy et al., 2021) as pseudo input, replacing the actual input data $\mathbf{P} + \mathbf{X}$ or intermidiate outputs. Remember that this only happens when we solve the initialization to avoid data-dependent initialization, while in the training stage the input is not changed. The ablation study of different pseudo inputs is presented in Appendix C.1.

With the use of impulse filters and the pseudo input, Eq. (6) becomes

$$\mathbf{M}_{\text{init}} = \text{softmax}(\mathbf{P}\mathbf{Q}_{\text{init}}\mathbf{K}_{\text{init}}^T\mathbf{P}^T) \approx \mathbf{H}_{\text{impulse}}. \tag{7}$$

## 4.2 Solving $\mathbf{Q}_{\text{INIT}}$ and $\mathbf{K}_{\text{INIT}}$

There exist numerous approaches to solve Eq. (7) for $\mathbf{Q}_{\text{init}}$ and $\mathbf{K}_{\text{init}}$ with known $\mathbf{H}_{\text{impulse}}$ and $\mathbf{P}$. In mimetic initialization (Trockman & Kolter, 2023), the product $\mathbf{Q}_{\text{init}}\mathbf{K}_{\text{init}}^T$ is initialized following a certain distribution. Singular value decomposition (SVD) is utilized to solve $\mathbf{Q}_{\text{init}}$ and $\mathbf{K}_{\text{init}}$. While a similar SVD-based approach could be employed in our scenario—despite we intend to initialize the softmax attention map, it is found to be ineffective due to the large error resulting from the pseudo-inverse of $\mathbf{P}$ and low-rank approximation. Consequently, we opt not to pursue an analytical solution but rather employ a simple optimization to obtain $\mathbf{Q}_{\text{init}}$ and $\mathbf{K}_{\text{init}}$. This approach also addresses concerns regarding scale and layer normalization in the attention mechanism.

The pseudo code for our initialization strategy is shown in Algorithm 1. In the first step, we compute the attention map $\mathbf{H}_{\text{impulse}}$ based on the 2D impulse convolution matrix. The pseudo input $\tilde{\mathbf{X}}$ is then computed through the absolute positional encoding $\mathbf{P}$. Note that the pseudo input $\tilde{\mathbf{X}}$ remains constant throughout the entire optimization process without requiring re-sampling. Additionally, the constant scale $\sigma$, and any normalization techniques such as layer normalization or batch normalization remain consistent with those utilized in ViT.

To optimize $\mathbf{Q}_{\text{init}}$ and $\mathbf{K}_{\text{init}}$, our objective function is defined as

$$\underset{\mathbf{Q}_{\text{init}}, \mathbf{K}_{\text{init}}}{\arg\min} \frac{1}{N^2} \left\| \mathbf{H}_{\text{impulse}} - \text{softmax}\left(\sigma\tilde{\mathbf{X}}\mathbf{Q}_{\text{init}}\mathbf{K}_{\text{init}}^T\tilde{\mathbf{X}}^T\right) \right\|_F^2, \tag{8}$$

---

**Algorithm 1** Convolutional structured impulse initialization for ViT

---

**Input:** $\mathbf{P}$, $f$                                  ▷ Positional encoding, convolution filter size
**Output:** $\mathbf{Q}_{\text{init}}$, $\mathbf{K}_{\text{init}}$                             ▷ Initialization of attention parameters
    $N, D \leftarrow$ shape of $\mathbf{P}$
    $\mathbf{H}_{\text{impulse}} \leftarrow ImpulseConvMatrix(N, f)$                ▷ Build 2D impulse convolution matrix
    $\tilde{\mathbf{X}} \leftarrow LayerNorm(\mathbf{P})$                                  ▷ Get pseudo input
    $\sigma \leftarrow \frac{1}{\sqrt{D/h}}$                                         ▷ Scale in attention
    $\mathbf{Q}_{\text{init}}, \mathbf{K}_{\text{init}} \leftarrow Parameters(\cdot)$               ▷ Random initialized before optimization
    **for** $i \leftarrow 1, max\_iter$ **do**
        $\hat{\mathbf{H}}_{\text{impulse}} \leftarrow \text{softmax}(\sigma \tilde{\mathbf{X}} \mathbf{Q}_{\text{init}} \mathbf{K}_{\text{init}}^T \tilde{\mathbf{X}}^T)$
        $Loss \leftarrow \text{MSE}(\hat{\mathbf{H}}_{\text{impulse}}, \mathbf{H}_{\text{impulse}})$
        Compute gradients and update $\mathbf{Q}_{\text{init}}$ and $\mathbf{K}_{\text{init}}$
    **end for**
    **return** $\mathbf{Q}_{\text{init}}$, $\mathbf{K}_{\text{init}}$

---

where $\tilde{\mathbf{X}}$ is the normalized pseudo input, and $\mathbf{Q}_{\text{init}}$ and $\mathbf{K}_{\text{init}}$ can be random initialized before optimization. We then optimize for $max\_iter = 10,000$ epochs using Adam optimizer (Kingma & Ba, 2015) with a learning rate of $1e^{-4}$ and mean squared error (MSE) loss. It is worth noting that this optimization is not a pretrained step since no real data is involved. Rather, our optimization algorithm serves as a surrogate for the SVD solver, converging in just a few seconds (∼5s).

## 5 EXPERIMENTS AND ANALYSIS

### 5.1 SETTINGS

**Dataset.** We evaluate our structured initialization strategy on the small-scale datasets CIFAR-10, CIFAR-100 (Krizhevsky et al., 2009), SVHN (Yuval, 2011) with $2 \times 2$ patches. Additionally, we test our model on a large-scale ImageNet-1K (Deng et al., 2009) dataset with $16 \times 16$ patches. Furthermore, in validating our theory on ConvMixer, we conduct all ConvMixer related experiments in Sec. 5.5 on CIFAR-10.

**Models.** Our experiments primarily focus on the tiny ViT model, namely ViT-T (Dosovitskiy et al., 2021). Instead of using the classification token and a learnable positional encoding as defined in ViT, we use the average global pooling and a sinusoidal absolute positional encoding. In general, these small tweaks will not compromise the performance of ViTs. On the contrary, as shown in Tab. 1, these two modifications lead to improved performance on the CIFAR-10 dataset. Henceforth, all the following experiments use this configuration. The default architecture of ViT-T includes a depth of 12, an embedding dimension of 192, and 3 heads. The default architecture of ViT-S includes a depth of 12, an embedding dimension of 384, and 6 heads.

**Training.** We utilize the PyTorch Image Models (timm) (Wightman, 2019) to train all ViT models. We employ a simple random augmentation strategy from (Cubuk et al., 2020) for data augmentation. Our models were trained with a batch size of 512 using the AdamW (Loshchilov & Hutter, 2019) optimizer, with a learning rate of $10^{-3}$ and weight decay set to 0.01, for 200 epochs. Note that all experiments were conducted on the Tesla V100 SXM3 with 32GB memory.

**Initialization.** Considering that the number of heads in ViTs is typically small, we utilized both $3 \times 3$ (Imp.-3) and $5 \times 5$ (Imp.-5) filters for our structured initialization method. We compare our method with Pytorch default initialization (Kaiming Uniform (He et al., 2015)), timm default initialization (Trunc Normal), and mimetic initialization (Mimetic (Trockman & Kolter, 2023)).

### 5.2 RESULTS ACROSS DATASETS

In Tab. 2, we present the results of five different methods across four datasets. For ImageNet-1K, we follow the training settings defined in ConvMixer (Trockman & Kolter, 2022), training all models for 300 epochs. Our proposed methods, both Imp.-3 and Imp.-5, demonstrate comparable—if not

Table 1: Classification accuracy(%) of ViT-T with various basic settings on CIFAR-10.

| Model | Classification Token | Average Pooling |
|---|---|---|
| Learnable PE | 81.23 | 82.23 |
| Sinusoidal PE | 83.17 | **85.30** |

Table 2: Classification accuracy(%) of ViT-T using different initialization methods on CIFAR-10, CIFAR-100, SVHN and ImageNet-1K. Red number indicates accuracy decrease, and green number indicates an increase in accuracy. Note that we compare the performance to the Trunc Normal initialization method (shaded in gray).

| Method | CIFAR-10 | CIFAR-100 | SVHN | ImageNet-1K |
|---|---|---|---|---|
| Kaiming Uniform (He et al., 2015) | 86.36 2.27↓ | 63.50 3.00↓ | 94.51 1.31↑ | 74.11 0.69↑ |
| Trunc Normal | 88.63 | 66.50 | 93.20 | 73.42 |
| Mimetic (Trockman & Kolter, 2023) | 91.16 2.53↑ | 70.40 3.90↑ | **97.53** 4.33↑ | 74.34 0.92↑ |
| Ours (Imp.-3) | **91.62** 2.99↑ | 68.81 2.31↑ | 97.21 4.01↑ | 74.24 0.82↑ |
| Ours (Imp.-5) | 90.67 2.04↑ | **70.46** 3.96↑ | 97.23 4.03↑ | **74.40** 0.98↑ |

superior—performance compared to mimetic initialization. Particularly on smaller-scale datasets like CIFAR-10, CIFAR-100, and SVHN, known to pose challenges for ViT models, our method consistently exhibits 2% to 4% improvement compared to Trunc Normal. Notably, our method maintains to perform well on large-scale datasets like ImageNet-1K, which shows that our structured initialization keeps the flexibility of the attention map even when learning from large-scale data.

## 5.3 RESULTS ACROSS MODELS

Although our method demonstrates impressive performance when training ViT-T on small-scale datasets, the model ViT-T only has 3 heads, which falls short of the requirements defined in Proposition 1. To better showcase the advantage of our method, we increased the number of heads to 8 in ViT-T, denoted as ViT-T/h8. In addition to the experiments with ViT-T, we also tested our method on the small ViT model (ViT-S). The configuration of ViT-S includes an embedding dimension of 384, a depth of 12, and 6 heads, denoted as ViT-S/h6. Furthermore, we increasde the number of heads to 16 and denoted this model as ViT-S/h16. The results on CIFAR-100 are shown in Tab. 3.

As the model size increases, particularly with a higher number of heads, our initialization method demonstrates improved and more stable performance, bringing a larger gap between other initialization methods. This performance increase proves our theory (see Proposition 1) regarding the expressibility of spatial mixing matrix: more heads provide more linearly independent filters. For instance, when the number of heads is 3, as in ViT-T/h3, each layer contains only 3 unique "filters" with each filter having $192 / 3 = 64$ copies. While the number of copies is sufficient, having only 3 unique "filters" is inadequate for forming the filter basis, even for a $3 \times 3$ random impulse filter.

As we increase the number of heads, we observe an adequate improvement in the performance of our method. However, maintaining a constant embedding dimension while increasing the number of heads leads to fewer copies per head. While this may not present a significant issue in ConvMxier as long as the number of copies exceeds the rank of the inputs, a notable challenge arises with multi-head attention: the dimensionality of $\mathbf{Q}$ and $\mathbf{K}$ will decrease to $D / h$ as the number of heads $h$ increases. Consequently, the rank of $\mathbf{Q}$ and $\mathbf{K}$ diminishes considerably, making it more challenging for the low-rank approximation $\mathbf{Q}\mathbf{K}^T$ to learn an effective attention map.

This phenomenon may explain why the Kaiming Uniform and Trunc Normal methods occasionally exhibit inferior performance as the number of heads increases. For the mimetic initialization, the situation is potentially more problematic, as it utilizes SVD to solve for a low-rank $\mathbf{Q}$ and $\mathbf{K}$. As the number of heads increases, resulting in a lower rank, the approximation error grows, further deviating the actual $\mathbf{Q}\mathbf{K}^T$ from the anticipated value. In contrast, our initialization strategy employs an iterative optimization method, which helps mitigate errors arising from low-rank approximations. Consequently, our method benefits more when applied with a larger number of heads.

Table 3: Classification accuracy(%) of ViT-T/h3, ViT-T/h8, ViT-S/h6 and ViT-S/h16 using different initialization methods on CIFAR-100. Red number indicates accuracy decrease, and green number indicates an increase in accuracy. Note that we compare the performance to the Trunc Normal initialization method (shaded in gray).

| Method | ViT-T/h3 | ViT-T/h8 | ViT-S/h6 | ViT-S/h16 |
|---|---|---|---|---|
| Kaiming Uniform (He et al., 2015) | 63.50 3.00↓ | 63.09 3.39↓ | 66.06 0.75↓ | 64.61 2.64↓ |
| Trunc Normal | 66.50 | 66.48 | 66.81 | 67.25 |
| Mimetic (Trockman & Kolter, 2023) | 70.40 3.90↑ | 69.93 3.45↑ | 73.86 7.05↑ | 72.72 5.47↑ |
| Ours (Imp.-3) | 68.81 2.31↑ | 70.79 4.31↑ | **75.97** 9.16↑ | **75.40** 8.15↑ |
| Ours (Imp.-5) | **70.46** 3.96↑ | **70.86** 4.38↑ | 73.49 6.68↑ | 74.27 7.02↑ |

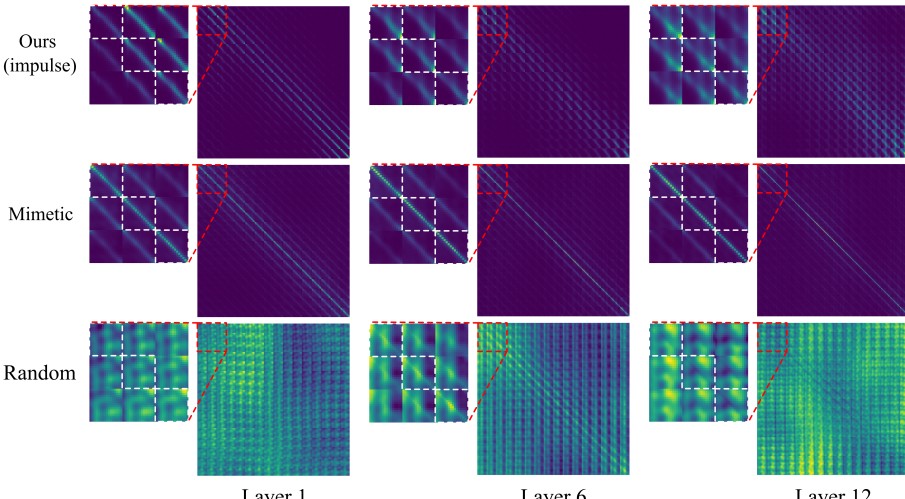

Layer 1        Layer 6        Layer 12

Figure 3: Visualization of attention maps in ViT-T using ours, mimetic (Trockman & Kolter, 2023), and random (Liu et al., 2022) initializations. Red boxes highlight zoomed-in details of the $48 \times 48$ upper left corner in attention maps. White boxes indicate the main diagonal blocks of the zoomed-in attention maps. Our structured initialization method offers off-diagonal attention peaks aligned with the impulse structures, whereas mimetic initialization primarily strengthens the main diagonal of the attention map. Random initialization shows little to no patterns.

## 5.4 ATTENTION MAPS

To show the effectiveness of our initialization on constraining the structure of attention maps, we show the averaged attention maps across CIFAR-10 training data using ViT-T/h3 at 1st, 6th, and 12th layers after initialization before training in Fig. 3. The attention map showed here does not have an exact pattern as seen in impulse filters since we only use positional encoding as pseudo inputs to optimize the initial attention parameters, while the real input to each attention layer is $\mathbf{P} + \mathbf{X}$ or intermediate outputs from the previous layer. Nevertheless, there exists a clear pattern of convolutional structures. Since mimetic initialization only focusing on distribution of weights, the structure of attention map can be merely strengthened along the diagonal. Our initialization method offers off-diagnoal peaks in convolutional structure in attention map. As the network layer goes deeper, these peaks become less visible, because the difference between pseudo input and real input is getting larger through layers. The visualization of attention maps for all the layers from 1 to 12 can be found in Appendix C.3

## 5.5 COMPARING VIT WITH CONVMIXER

**Experimental evidence for proposition and corollaries in Sec. 3.** To validate our findings discussed in Sec. 3 regarding the effectiveness of random filters, we train ConvMixer (Trockman & Kolter, 2022) models with an embedding dimension of 256, a depth of 8, and a patch size of 2 on the CIFAR-10 dataset, using filter size of 3, 5, and 8. We followed the same configurations defined in Sec. 5.1, except for setting the learning rate to 0.01, and the number of epochs to 100. Additionally, we tested with a ConvMixer with an embedding dimension of 512. The results are shown in Tab. 4.

Table 4: Classification accuracy(%) of ConvMixer (depth 8) with different filter sizes, embedding dimensions on CIFAR-10.

| Kernel Size | Embedding Dimension = 256 | | | | Embedding Dimension = 512 | | | |
|---|---|---|---|---|---|---|---|---|
| | Trained | Random | Impulse | Box | Trained | Random | Impulse | Box |
| 3 | 91.76 | 90.72 | 90.68 | 81.70 | 92.82 | 92.15 | 92.20 | 81.90 |
| 5 | 92.69 | 90.87 | 90.41 | 80.57 | 93.90 | 92.72 | 91.91 | 81.19 |
| 8 | 92.34 | 88.12 | 87.82 | 78.95 | 92.96 | 90.09 | 89.61 | 80.10 |

Table 5: Classification accuracy(%) of ViT (depth 8) with a different number of heads, embedding dimensions on CIFAR-10.

| Method | Embedding Dimension = 256 | | | | Embedding Dimension = 512 | | | |
|---|---|---|---|---|---|---|---|---|
| | h4 | h8 | h16 | h32 | h4 | h8 | h16 | h32 |
| Kaiming Uniform (He et al., 2015) | 85.71 | 85.16 | 84.62 | 84.24 | 87.28 | 86.50 | 85.07 | 84.17 |
| Trunc Normal | 87.27 | 87.10 | 87.30 | 86.71 | 87.49 | 87.03 | 87.74 | 87.39 |
| Mimetic (Trockman & Kolter, 2023) | **90.52** | 89.45 | 88.97 | 86.83 | 90.94 | 90.75 | 90.35 | 89.27 |
| Ours (Imp.-3) | 89.95 | **90.67** | 90.59 | 88.69 | **91.55** | **91.75** | 91.49 | **91.18** |
| Ours (Imp.-5) | 90.08 | 90.38 | **90.61** | **89.14** | 90.92 | 91.67 | **91.87** | 90.84 |

We tested on the end-to-end trained ConvMixer along with three different initialization methods: random (Corollary 1), impulse (Corollary 2), and box (Corollary 3). Please note that the three initialization methods only initialize the spatial convolution filters without training. Specifically, the box filters use all ones, serving as an average pooling function.

In general, random and impulse initialization achieve comparable accuracy compared to the end-to-end trained model, while box initialization exhibits inferior performance. This discrepancy can be attributed to the deficient rank of box filters, as they lack $f^2$ linearly independent filters, unlike random and impulse initialization, which can form the basis of the filter space.

**ViT and ConvMixer of same embedding dimension and depth.** For ConvMixers with the same embedding dimension, the performance gap between trained filters and random or impulse widens as the kernel size increases. As we discussed in Sec. 5.3, as the kernel size increases, more unique filters are needed to form the basis of filter space. Consequently, each head (unique filter) has fewer copies, making it difficult to match the rank of inputs, thus failing to meet the condition in Proposition 1. When the embedding dimension doubles, the performance gap between trained filters and random or impulse filters diminishes with the same kernel size. However, models with larger kernel sizes still tend to have a bigger gap due to an insufficient number of copies for each unique filter.

Our method is motivated by the similarity between ViT and ConvMixer. To show this connection, we train ViTs with similar configurations to ConvMixer as described in Sec. 5.5. Specifically, we train ViTs of a depth of 8 with embedding dimensions of 256 and 512. The number of heads is from 4 to 32. Results of different initialization methods are shown in Tab. 5. We also provide additional results with embedding dimensions of 64 and 512 in Appendix C.2

Our impulse initialization methods demonstrate superior performance across nearly all configurations. Especially, our method achieves a top accuracy of 90.67% and 91.87% with an embedding dimension of 256 and 512, respectively, significantly outperforming other initialization methods. Moreover, our method achieves results on par with end-to-end trained ConvMixers (91.76% and 92.82%) of equal depth and embedding dimensions. In addition, these results with different numbers of heads validate our discussions in Sec. 5.3 regarding the impact of the number of heads on the performance.

# 6 CONCLUSION

In this paper, we propose a structured initialization method with convolutional impulse filters for attention maps in ViTs. Our method preserves both the advantage of locality within CNNs and the capacity to learn long-range dependencies inherited from ViTs. We also provide a thorough theoretical explanation of the spatial and channel mixing in ConvMixer and ViT, building connections between the structural bias in CNNs and the initialization of ViTs. Our results on small-scale datasets validate the effectiveness of the convolutional structural bias, while on-par performance on large-scale datasets indicates the preservation of architectural flexibility.

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

## A    PROOF FOR PROPOSITION 1

Let $\mathbf{w} = [w_1, w_2, \ldots, w_D]^T \in \mathbb{R}^{D \times 1}$ be the channel mixing weights for one output channel and $\mathbf{H}_1, \mathbf{H}_2, \ldots, \mathbf{H}_D$ are the corresponding spatial convolution filters for each channel. Therefore, the result $\mathbf{y} \in \mathbb{R}^N$ after spatial and channel mixing can be represented as,

$$\mathbf{y} = \sum_{i=1}^{D} w_i \mathbf{H}_i \mathbf{x}_i, \tag{9}$$

With Remark 1, we can suppose the rank of $\mathbf{X} \approx \mathbf{Z}\mathbf{A}$ is $k$, where $\mathbf{Z} = [\mathbf{z}_1, \ldots, z_k]$ and $k \ll D$, as illustrated in Fig. 2. We then obtain

$$\mathbf{y} \approx \sum_{i=1}^{D} \sum_{j=1}^{k} w_i a_{ji} \mathbf{H}_i \mathbf{z}_j = \sum_{j=1}^{k} \tilde{\mathbf{H}}_j \mathbf{z}_j, \tag{10}$$

where $a_{ji}$ refers to the row $j$, column $i$ element of $\mathbf{A}$, and $\tilde{\mathbf{H}}_j = \sum_{i=1}^{D} w_i\, a_{ji}\, \mathbf{H}_i$.

Remember that a linear combination of $f^2$ linearly independent filters $\mathbf{h}$ can express any arbitrary filter in filter space $\mathbb{R}^{f \times f}$, where $\mathbf{h}$ serves as the basis. Consequently, any desired $\tilde{\mathbf{H}}_1, \tilde{\mathbf{H}}_2, \ldots, \tilde{\mathbf{H}}_D$ can be achieved by only learning the channel mixing weights $\mathbf{w}$. Therefore, we obtain the following proposition.

## B    CONVOLUTIONAL REPRESETATION MATRIX

In Sec. 3, we interchangeably use the terms convolution filter $\mathbf{h}$ and convolution matrix $\mathbf{H}$. Additionally, we represent the impulse filter as a convolutional matrix. Here, we offer a detailed explanation of the relationship between the convolutional filters and the convolutional matrices.

Let us define a 2D convolution filter as $\mathbf{h} \in \mathbb{R}^{f \times f}$ with elements

$$\mathbf{h} = \begin{pmatrix} h_{11} & \cdots & h_{1f} \\ \vdots & \ddots & \vdots \\ h_{f1} & \cdots & h_{ff} \end{pmatrix}. \tag{11}$$

When $\mathbf{h}$ is convolved with an image $\mathbf{x} \in \mathbb{R}^{H \times W}$, this convolution operation is equivalent to a matrix multiplication

$$\mathrm{vec}(\mathbf{h} * \mathbf{x}) = \mathbf{H}\,\mathrm{vec}(\mathbf{x}), \tag{12}$$

where $\mathbf{H}$ is composed from the elements in $\mathbf{h}$ and zeros in the following format:

$$\mathbf{H} = \begin{pmatrix} \mathbf{F_1} & \mathbf{F_2} & \cdots & \mathbf{F_f} & \mathbf{0} & \mathbf{0} & \cdots & \mathbf{0} \\ \mathbf{0} & \mathbf{F_1} & \mathbf{F_2} & \cdots & \mathbf{F_f} & \mathbf{0} & \cdots & \mathbf{0} \\ \vdots & \ddots & \ddots & \ddots & \ddots & \ddots & \ddots & \vdots \\ \mathbf{0} & \cdots & \mathbf{0} & \mathbf{F_1} & \mathbf{F_2} & \cdots & \mathbf{F_f} & \mathbf{0} \\ \mathbf{0} & \cdots & \mathbf{0} & \mathbf{0} & \mathbf{F_1} & \mathbf{F_2} & \cdots & \mathbf{F_f} \end{pmatrix}, \tag{13}$$

where

$$\mathbf{F_i} = \begin{pmatrix} h_{i1} & h_{i2} & \cdots & h_{if} & 0 & 0 & \cdots & 0 \\ 0 & h_{i1} & h_{i2} & \cdots & h_{if} & 0 & \cdots & 0 \\ \vdots & \ddots & \ddots & \ddots & \ddots & \ddots & \ddots & \vdots \\ 0 & \cdots & 0 & h_{i1} & h_{i2} & \cdots & h_{if} & 0 \\ 0 & \cdots & 0 & 0 & h_{i1} & h_{i2} & \cdots & h_{if} \end{pmatrix}, \tag{14}$$

for $i = 1, 2, \ldots, f$. $\mathbf{F_i}$s are circulant matrices and $\mathbf{H}$ is a block circulant matrix with circulant block (BCCB). Note that convolutions may employ various padding strategies, but the circulant structure remains consistent. Here, we show the convolution matrix without any padding as an example.

Table 6: Classification accuracy(%) of different pseudo input on CIFAR-10. Green shaded row indicates the best choice of pseudo inputs when achieving the best average accuracy. "PE" denotes positional encoding, and "G" represents random sampling from the Gaussian distribution. "Imp.-3" represents a $3\times3$ impulse filter, and "Imp.-5" denotes a $5\times5$ impulse filter.

| Pseudo Input | | Same $\mathbf{Q}_{init}$ and $\mathbf{K}_{init}$ | | | | Different $\mathbf{Q}_{init}$ and $\mathbf{K}_{init}$ | | | | Avg. |
| | | ViT-T/h3 | | ViT-T/h8 | | ViT-T/h3 | | ViT-T/h8 | | |
| First Layer | Following Layers | Imp.-3 | Imp.-5 | Imp.-3 | Imp.-5 | Imp.-3 | Imp.-5 | Imp.-3 | Imp.-5 | |
|---|---|---|---|---|---|---|---|---|---|---|
| PE | PE | 90.75 | 90.22 | 90.39 | 90.24 | 89.90 | 90.18 | 90.19 | 91.24 | **90.39** |
| PE | U | 86.90 | 86.10 | 87.99 | 87.86 | 87.35 | 85.56 | 86.40 | 86.61 | 86.85 |
| PE | PE+U | 89.62 | 88.40 | 89.61 | 89.50 | 88.99 | 89.29 | 89.05 | 89.47 | 89.24 |
| U | PE | 90.34 | 89.21 | 90.96 | 89.83 | 90.76 | 90.00 | 91.20 | 90.34 | 90.33 |
| U | U | 86.05 | 86.50 | 86.13 | 86.44 | 87.09 | 87.22 | 86.03 | 86.13 | 86.45 |
| U | PE+U | 89.91 | 89.99 | 89.96 | 89.93 | 89.94 | 89.89 | 89.60 | 89.07 | 89.79 |
| PE+U | PE | 90.07 | 89.19 | 90.31 | 89.56 | 90.03 | 90.72 | 90.40 | 90.76 | 90.13 |
| PE+U | U | 86.07 | 86.52 | 85.55 | 85.94 | 86.02 | 86.07 | 86.52 | 85.55 | 86.02 |
| PE+U | PE+U | 89.63 | 89.33 | 90.02 | 88.92 | 89.52 | 89.28 | 89.29 | 89.33 | 89.41 |

Table 7: Classification accuracy(%) of ViT (depth 8) with a different number of heads, embedding dimensions on CIFAR-10.

| Method | Embedding Dimension = 64 | | | | Embedding Dimension = 128 | | | |
| | h4 | h8 | h16 | h32 | h4 | h8 | h16 | h32 |
|---|---|---|---|---|---|---|---|---|
| Kaiming Uniform (He et al., 2015) | 81.26 | 80.76 | 80.71 | 79.89 | 84.46 | 83.83 | 83.05 | 82.70 |
| Trunc Normal | 81.57 | 82.12 | 81.57 | **80.40** | 86.19 | 86.00 | 85.39 | 84.37 |
| Mimetic (Trockman & Kolter, 2023) | **84.59** | 82.78 | **81.78** | 79.71 | 87.90 | 87.49 | 85.51 | 83.68 |
| Ours (Imp.-3) | 82.73 | **84.12** | 81.59 | 79.39 | **88.49** | 87.75 | **87.63** | 84.42 |
| Ours (Imp.-5) | 83.43 | 82.66 | 80.80 | 79.11 | 87.82 | **88.02** | 87.57 | **84.79** |

# C  ADDITIONAL RESULTS

## C.1  PSEUDO INPUT

We offer additional results for the ablation study on pseudo input with random inputs sampled from a Uniform distribution in Tab. 6. The trend is aligned with the findings in the main paper for random inputs sampled from a Gaussian distribution. However, the performance of using random inputs sampled from a Uniform distribution is worse than those using a Gaussian distribution, since the real input data is more likely to follow a Gaussian distribution.

## C.2  RELATIONSHIP BETWEEN HEAD NUMBERS AND EMBEDDING DIMENSION

In the main paper, we have discussed the relationship between the number of heads and the embedding dimensions. We have provided the results with embedding dimensions of 256 and 512 for a ViT with depth 8. Here we show additional results with embedding dimensions of 64 and 128 in Tab. 7. The trend is consistent with the discussions in the main paper.

## C.3  ATTENTION MAPS

Here we provide additional visualization of the attention maps for all 12 layers in Fig. 4 and Fig. 5. Additionally, we have provided the attention maps before and after training in Fig. 6 to further demonstrate the advantage of our method in capturing both local and global dependencies.

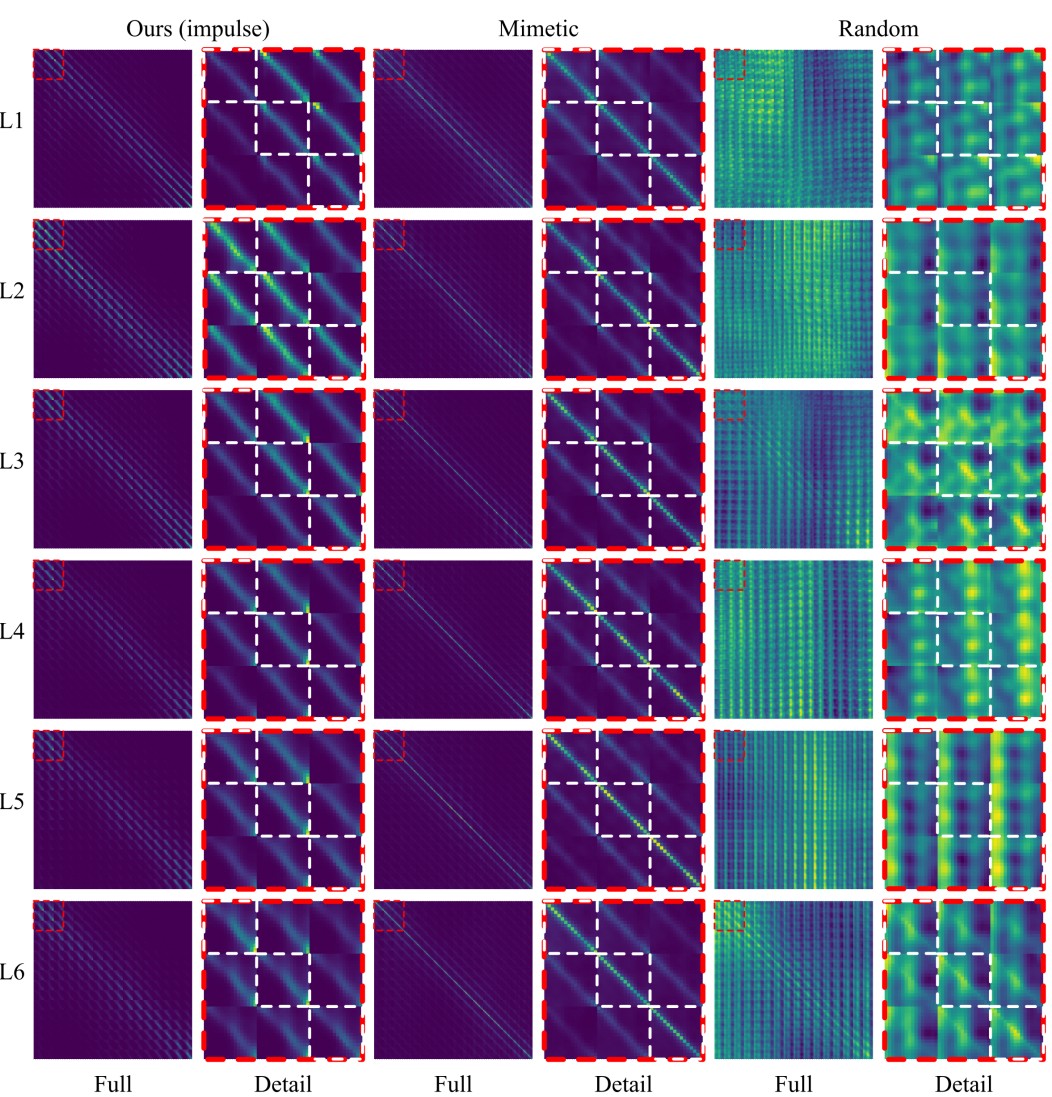

Figure 4: Visualization of attention maps in ViT-T using our impulse initialization method, mimetic (Trockman & Kolter, 2023), and random (Liu et al., 2022) initializations. Red boxes highlight zoomed-in details of the $48 \times 48$ upper left corner in attention maps. White boxes indicate the main diagonal blocks of the zoomed-in attention maps. Our structured initialization method offers off-diagonal attention peaks aligned with the impulse structures, whereas mimetic initialization primarily strengthens the main diagonal of the attention map. Random initialization shows little to no patterns.

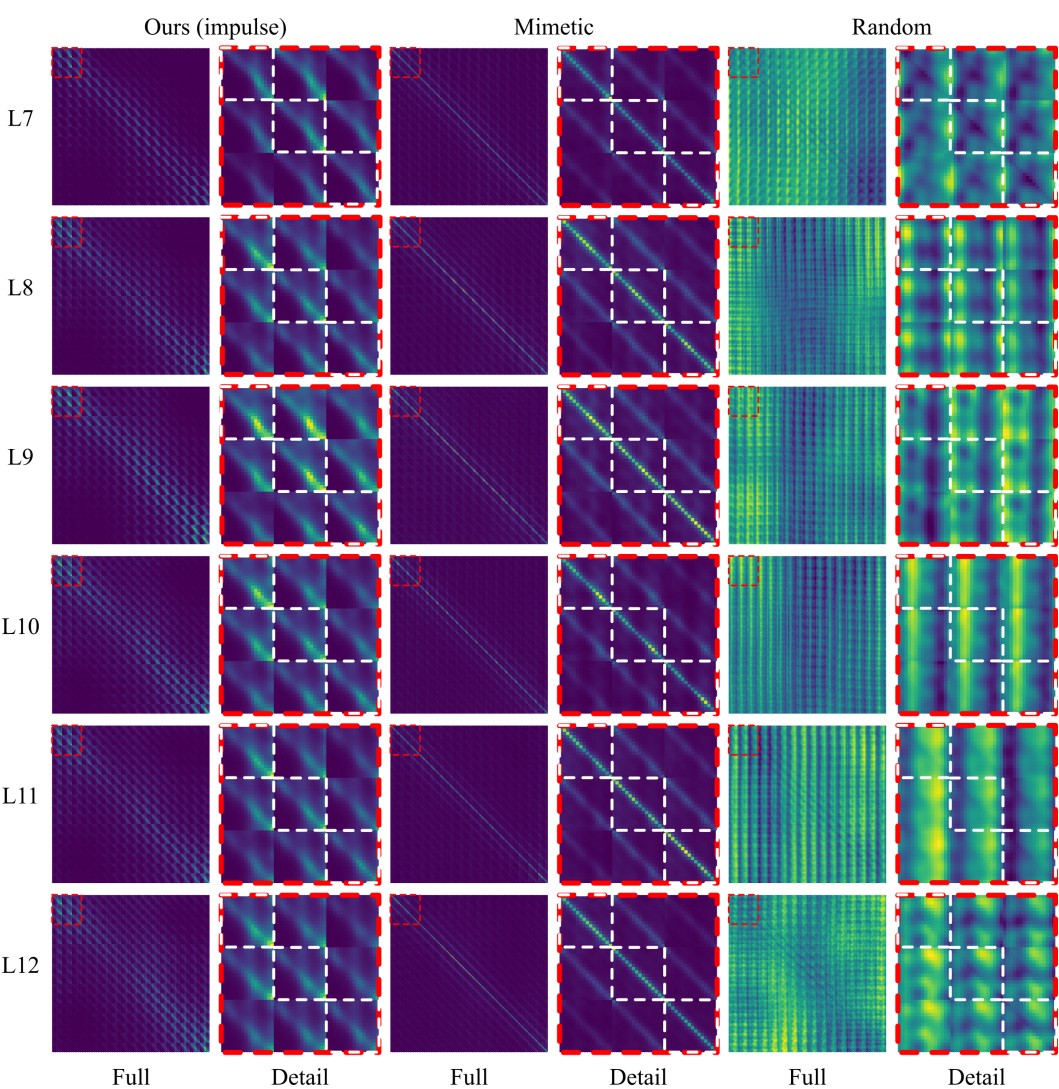

Figure 5: Visualization of attention maps in ViT-T using our impulse initialization method, mimetic (Trockman & Kolter, 2023), and random (Liu et al., 2022) initializations. Red boxes highlight zoomed-in details of the $48 \times 48$ upper left corner in attention maps. White boxes indicate the main diagonal blocks of the zoomed-in attention maps. Our structured initialization method offers off-diagonal attention peaks aligned with the impulse structures, whereas mimetic initialization primarily strengthens the main diagonal of the attention map. Random initialization shows little to no patterns.

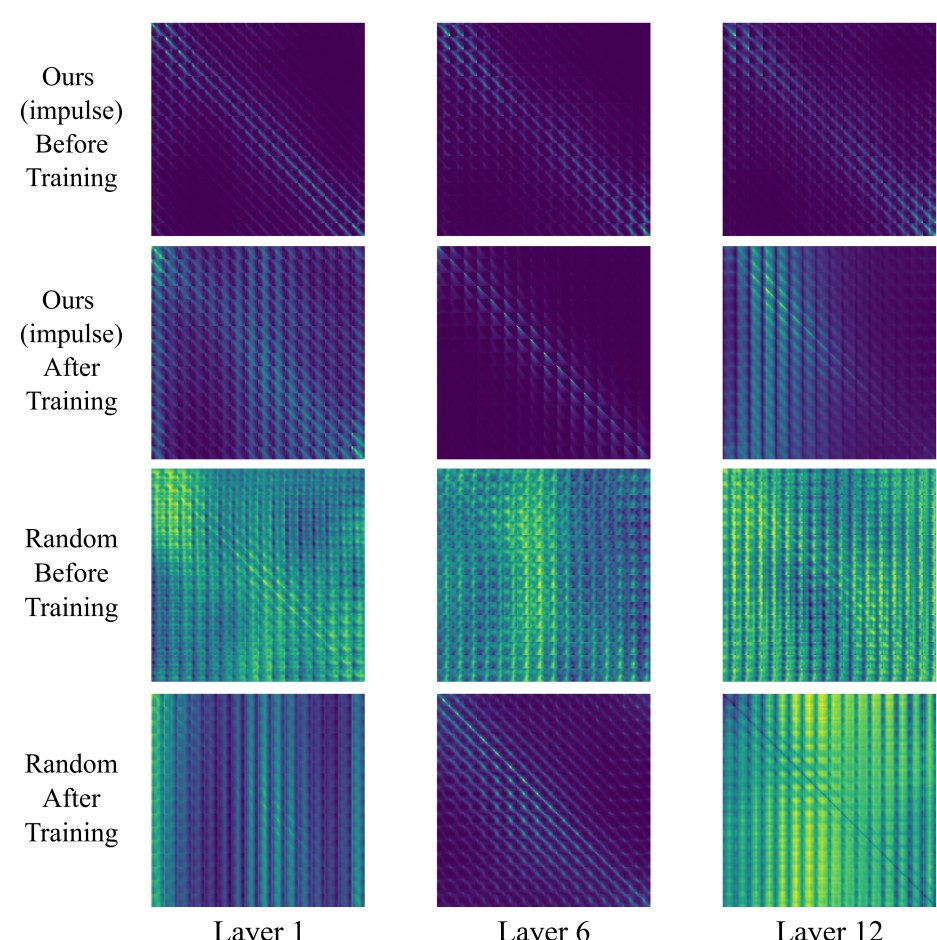

Layer 1          Layer 6          Layer 12

Figure 6: Visualization of attention maps in ViT-T using our impulse initialization method and conventional random initialization on the CIFAR-10 dataset before and after training. Our structured initialization method offers off-diagonal attention peaks aligned with the impulse structures during initialization, providing the necessary locality information, and helping with training on small-scale datasets. On the other hand, random initialization has no structured attention map during initialization, making training harder for small-scale datasets. Despite being initialized using the impulse structures, our method can still capture long-range, global dependencies after training, since our method does not change the Transformer architectures in ViTs.

