# OpenReview forum: "Structured Initialization for Attention in Vision Transformers"
_ICLR.cc/2025/Conference — Submitted to ICLR 2025_

### Official Review · Reviewer_tf2y · 2024-10-21

**Soundness:** 2
**Presentation:** 3
**Contribution:** 2
**Rating:** 3
**Confidence:** 5

**Summary:**

This paper introduces a novel method for improving the performance of ViTs when trained on small-scale datasets by incorporating structured initialization. The authors identify that ViTs struggle with small datasets compared to CNNs, which benefit from inherent inductive biases. By reinterpreting the architectural bias in CNNs as an initialization bias for ViTs, the authors propose a "structured initialization" method that results in structured attention maps for ViTs. The key contribution lies in the use of random convolutional impulse filters to guide the initialization process. The method is theoretically justified and empirically validated across serveral benchmarks. The paper demonstrates that structured initialization yields performance improvements on small datasets without compromising ViT’s flexibility on larger datasets.

**Strengths:**

Structured architecture achieves good performance across both small and large datasets, which demonstrates its scalability and flexibility.

**Weaknesses:**

1. The core argument of the method is that the convolutional structure can be transferred to the attention mechanism in transformers by initializing the attention maps with random impulse filters. However, this analogy between convolutional layers in CNNs and the attention mechanism in ViTs may be overly simplistic. CNNs' convolutional filters are spatially local and fixed in structure, while attention in ViTs is meant to capture long-range dependencies and is more flexible. This difference is crucial, and the method does not seem to fully address how imposing a rigid, convolution-like structure at initialization aligns with the flexibility that the attention mechanism needs. The convolution structure might limit the model's ability to learn long-range dependencies that are essential to the transformer. The claim that random impulse filters can replace learned convolutional filters is somewhat true for CNNs under certain conditions (like ConvMixers), but applying this to ViTs is more challenging. The attention mechanism is a more complex and dynamic operation compared to convolutions, and it’s unclear if the same approximation can hold. In practice, imposing a convolution-like structure might hinder the attention mechanism's ability to adapt during training.

2. The paper proposes that impulse filters, combined with the softmax operation, can initialize the attention maps. The softmax function ensures that all outputs are non-negative, which is a crucial difference from convolutions, which can have both positive and negative values. Random convolutional filters may contain both positive and negative values, while softmax output does not. This inconsistency could cause issues. The authors acknowledge this (stating the filters must be positive), but they do not provide a deep exploration of how this might affect the quality or flexibility of the learned attention maps. Relying on impulse filters could reduce the model's expressivity, especially if the initialized filters are too rigid and only positive-valued patterns are learned initially.

3. The authors propose an iterative optimization process to solve for the initial values of $Q_{init}$ and $K_{init}$ such that the resulting attention maps resemble impulse convolution filters. The optimization is based on a pseudo-input, which is generated from positional encodings rather than actual data. This could introduce an unwanted bias into the model's initial learning process. While using positional encoding as pseudo-input is an interesting idea, the paper does not adequately explore how different choices of pseudo-inputs affect the results or whether using actual training data for initialization would be a better alternative.

4. The optimization process described is more computationally expensive (up to 10,000 iterations using Adam) compared with traditional initialization methods. This added complexity raises the question of whether the benefits of structured initialization outweigh the cost, especially given that the improvements on large datasets are marginal. There is no discussion of the computational cost vs. benefit of this method compared to standard initialization techniques.

5. The use of random impulse convolution filters assumes that locality is always important, but this assumption may not hold in tasks where global context is critical. In CNNs, locality is useful because of the hierarchical structure of learned features. However, in transformers, the attention mechanism is specifically designed to handle long-range dependencies. By forcing the model to start with local dependencies (via impulse filters), the authors may inadvertently restrict the model's ability to learn global features early in training, leading to potential issues in tasks where global context is key from the beginning.

6. The method relies on several hyperparameters, including filter size (3x3 or 5x5) and the number of iterations for optimization. However, the choice of these hyperparameters is not adequately justified or explored. The method's performance is likely sensitive to these parameters, but there is no thorough analysis of how variations in filter size or optimization parameters affect results. Given the complexity of the initialization process, these aspects should have been investigated in detail to ensure the method’s robustness.

**Questions:**

1. Why Impulse Filters? The reasoning behind choosing impulse filters (instead of other structured filters) for initializing attention maps could be explained in more detail. Are impulse filters the best possible choice, or could other filter types provide better generalization?

2. How does this structured initialization affect the deeper layers of ViTs after finetuning? Figure 3 is quite interesting. Do the constraints imposed by impulse filters affect the deeper layers’ ability to fine-tune long-range dependencies? The paper does not discuss the long-term effects of this initialization on the network’s convergence.

---

> ### Author Response · Authors · 2024-11-19
> **Response to reviewer tf2y**
>
> We thank the reviewer for acknowledging our method *“achieves good performance across both small and large datasets, which demonstrates its scalability and flexibility”*. We would like to clarify that the main questions the reviewer raised are the questions we want to address in the paper. Most of the questions the reviewers raised have been stated in the paper or solved in our method. We will answer all your concerns in detail.
>
> **Q: Introducing inductive bias in CNNs seems contradict to the ViT structures. How to do this? And how to approximate inductive bias in CNNs using random impulse filters?**
>
> **A:** We kindly remind the reviewer that the questions you have raised are exactly what we want to address in this paper. As convolution filters capture local and fixed patterns, which is good at learning on small-scale data but has limited learning ability to capture long-range dependencies, ViTs are good at learning long-range, dynamic global relationships. We argue that since the attention weights of conventional ViTs are randomly initialized, the performance of ViTs on small-scale datasets is limited. Therefore, we design a new initialization to introduce the inductive bias in CNNs to aid ViT training on small-scale datasets while **maintaining the Transformer architectures**.
>
> As also addressed in the common response, we would like to clarify that our method focuses on introducing the **architectural inductive bias** from CNNs to initialize the attention map **without changing the Transformer architectures**. We have emphasized this argument in the abstract and introduction, stating that, unlike previous arts that directly introduce convolutions into attention (which is also referred to by the reviewer as a more straightforward solution), damaging the structural advantages of Transformers, our method only introduces architectural inductive bias in attention map initialization, **maintaining the inherent structural flexibility in ViTs**. Therefore, our method **preserves the Transformer architectures** and still allows the ViTs to learn flexible, dynamic global relationships. This is also one of the central innovations of our method. In addition, we have also provided experiments on large-scale applications to validate the stable performance of our method that preserves the architectural flexibility of ViTs.
>
>
> **Q: Using impulse filters may reduce the model's expressivity. How is the performance of the proposed method using impulse filters to do initialization?**
>
> **A:** We want to refer the reviewer to our main paper and Appendix for the detailed performance of our method – both our theoretical and empirical results show that **random impulse filters (which are all positive)** perform equally well as the random convolutional filters and learned filters. And this particular design is one of the central innovations of our method. The concern that the reviewer raised is well-answered in our experiments – **using impulse filters to do initialization will not affect the expressivity of the model, and our method achieves good performance on both small-scale and large-scale datasets**.
>
>
> **Q: More experiments about different pseudo-inputs? Will using positional encoding introduce unwanted bias?**
>
> **A:** We kindly remind the reviewer that we have shown the results of various pseudo-inputs in Appendix Table 6. In our experiment, we have validated the effectiveness of using positional encoding as a pseudo-input. In addition, this pre-optimization step involves no real data, which is another contribution of our method being no need for pre-training. We are not sure what unwanted bias the reviewer referred to with positional encodings since **positional encodings are widely adopted in ViTs**.
>
>
> **Q: Is computational cost or attention map initialization an overhead?**
>
> **A:** We kindly refer the reviewer to Section 4.2 Line 289-291, where we stated that our pre-optimization is **not a pretrained step since no real data is involved**. Also, our optimization algorithm is very simple, and it serves as a surrogate for the SVD solver, converging in just a few seconds, *i.e.*, **around 5 seconds**. In addition, it is worth noting that for each embedding dimension and kernel size, the initialization only needs to be calculated once. Except for adding this pre-optimization, which only takes around 5 seconds, all the other training steps stay the same with conventional ViTs. Therefore, no additional cost overhead is introduced in our method.

---

> > ### Author Response · Authors · 2024-11-19
> > **Continued response to reviewer rf2y due to word limit**
> >
> > --- ***continued response*** ---
> >
> > **Q: Is introducing locality helpful or harmful?**
> >
> > **A:** We want to clarify that our initialization method is specifically designed for ViTs (vision tasks) where **locality matters**. In addition, we want to refer the reviewer to the first answer where we emphasized our method does not change the Transformer architecture, thus preserving the advantages of the Transformer in capturing dynamic, global long-range dependencies.
> >
> >
> > **Q: Is the method sensitive to hyperparameters? Experiments on different hyperparameter choices?**
> >
> > **A:** We want to clarify that our initialization method is **not sensitive to the hyperparameter choices**. As we stated in Section 4.2 and the previous answer, our pre-optimization of the attention map is only a surrogate for the SVD solver, it is designed to be a simple regression to speed up the SVD process. The number of iterations of the pre-optimization does not matter since this optimization will converge in a fast manner. For different kernel sizes, we have provided ablation studies and discussions in Line 358-364 and Table 4. In general, our method **only modifies the initialization of the attention map**, therefore, these hyperparameters do not play an important role since the model still learns from the training data after the initialization.
> >
> >
> > **Q: Why impulse filters?**
> >
> > **A:** We kindly refer the reviewer to a paragraph in Section 4.1 Line 230-238: ***why using impulse filters?*** and Proposition 1 for detailed explanations. From CNN's perspective, based on our theoretical analysis, impulse, random, and learned filters can perform equally well. From ViT's perspective, the softmax in attention determines that it can be only positive and tends to concentrate in one element in each row. In conclusion, when initializing a softmax attention map, **the most straightforward and suitable choice is random impulse convolution filters**.
> >
> > **Q: How does this structured initialization affect the deeper layers of ViTs after finetuning?**
> >
> > **A:** We want to clearly state that our method is **not a pre-training or fine-tuning method**. Our structured initialization does not include any form of pre-training on any dataset. We would like to remind the reviewer that Figure 3 has already shown the visualization of attention maps for layers 1, 6, and 12. We also provided a visualization of attention maps of all layers in Appendix Figure 4 and Figure 5. As we stated in the contribution and previous answers, our method only introduces inductive bias of CNNs in initialization without any changes to the Transformer architectures. Therefore, our method enjoys the advantages of both CNNs in capturing locality features and Transformers in capturing long-range dependencies.

---

> ### Author Response · Authors · 2024-11-21
> **Do you have further questions?**
>
> Dear reviewer tf2y, we have provided the rebuttal below. Did our response resolve all your questions? Since the discussion period is approaching, if you have further questions, please don't hesitate to ask. We are happy to resolve your concerns.

---

> > ### Comment · Reviewer_tf2y · 2024-11-23
> >
> > Thank you for the rebuttal but the response can not convince me at this moment, and I still have follow-up questions.
> >
> > 1. The response mentions that incorporating the inductive biases from CNNs aids ViT training while retaining the core Transformer architecture and its ability to learn global relationships. It's fair, but it seems to overlook potential trade-offs. Inductive biases like locality and translational invariance, inherent to CNNs, might constrain ViTs' flexibility in capturing global dependencies, particularly during initialization. Although the architecture remains unchanged, these biases could influence the model's training trajectory, potentially limiting its capacity to generalize to datasets where long-range dependencies dominate.
> >
> > 2. Impulse filters inherently lack the expressiveness of learned filters or even random convolutional filters, which are capable of capturing more diverse patterns. While the paper shows that impulse filters achieve similar results in specific cases, it does not sufficiently address whether this equivalence holds universally across tasks and datasets or why impulse filters would not constrain the model's expressivity.
> >
> > 3. Positional encodings impose a fixed structure on the model, which might interfere with its ability to adapt dynamically to certain tasks. The paper does not address whether this fixed inductive bias could be detrimental in scenarios where positional information conflicts with learned representations. For example, in tasks requiring permutation invariance, fixed positional encodings might hinder the model's performance.
> >
> > 4. The speed benefits of structured initialization are noted and appreciated.
> >
> > 5. I remain concerned about oversimplifying the effects of introducing locality. Even though the architecture is unchanged, the structured initialization creates a bias toward locality, which might influence how attention mechanisms prioritize global dependencies. The paper does not address whether this locality bias could inadvertently reduce the effectiveness of the global receptive field, which is a fundamental strength of Transformers.
> >
> > 6. The method's robustness to hyperparameters is a strong point.
> >
> > 7. While softmax attention maps often concentrate on a single element per row, this does not inherently validate the use of impulse filters over learned filters. The claim that impulse, random, and learned filters "perform equally well" lacks comprehensive empirical evidence in this specific context. The theoretical adaptability of learned filters suggests they could offer more benefits.
> >
> > 8. I saw these figures and thought those were visualized just after initialization. Am I correct? I am curious about the persistence or evolution of CNN-like inductive biases post-training. They will provide valuable insight into how the initialization might influence model behavior over time.

---

> > > ### Author Response · Authors · 2024-11-24
> > > **Further clarifications for Reviewer tf2y**
> > >
> > > We thank the reviewer for providing further feedback. And we are happy that we have resolved questions 4 and 6 in our previous comments. We will give further explanations.
> > >
> > > **A1**: To further explain “our method focuses on introducing the **architectural inductive bias** from CNNs to initialize the attention map **without changing the Transformer architectures**”, we have provided the attention maps of random and our structured initialization methods before and after training on the CIFAR10 dataset in **the revised paper Appendix Figure 6**. The caption of the figure is marked in red text. In a word, although being initialized using the convolutional structures, after training, our method can still capture global dependencies since **the ViT structure is not changed** in our method. We have also further emphasized this argument in the introduction and contribution sections.
> > >
> > > **A2**: We kindly refer the reviewer to Section 3 of our paper, where we theoretically show that for **the spatial mixing part**, impulse, random, and trained kernels are **identical**. In a word, any filters, learned or random, can be expressed as a linear combination of impulse filters, and the coefficients of the filters can be learned through the **channel mixing part**. In addition, we have provided experimental results in Table 4 which sufficiently shows that impulse, random, and trained spatial filters are equivalent in performance. This is a central innovation of our method to introduce the inductive bias of CNNs using the impulse filters.
> > >
> > > **A3**: We would like to refer the reviewer to Appendix Table 8 of the original ViT paper [c1]. In this table, the authors of [c1] have already shown that as long as positional encoding offers the position information, there is little difference in the ways to implement the positional encoding (at least for the vision task). Sinusoidal-based positional encoding is a widely used positional encoding, and it is suitable for our initialization method. Therefore, we chose the sinusoidal-based positional encoding for our ViT settings.
> > >
> > > **A5**: We would like to remind the reviewer that we have already provided experimental results on the large-scale dataset ImageNet-1K and show that the performance of ViT with our initialization method **performs equally well** as ViT using conventional initialization. In addition, introducing the inductive bias of CNNs into the initialization of ViTs helps train ViT on small-scale datasets, which is an unsolved issue of applying ViTs for small-scale vision tasks. We believe the strong performance of our methods on **both small-scale and large-scale datasets** validates the effectiveness of our method and the advantages of having locality information while not harming the global dependency information.
> > >
> > > **A7**: We refer the reviewer to **A2** for further details.
> > >
> > > **A8**: We would like to remind the reviewer that Figures 3, 4, and 5 are **attention maps during initialization**. The inductive bias of CNNs is only introduced during initialization to help with training. The training (optimization) will still allow the ViT to capture global dependency information since there are no architectural changes in ViTs. Please also see our revised paper Appendix Figure 6 for further details.
> > >
> > >
> > > **Reference**:
> > >
> > > [c1] Dosovitskiy, Alexey. "An image is worth 16x16 words: Transformers for image recognition at scale." arXiv preprint arXiv:2010.11929 (2020).

---

### Official Review · Reviewer_aWu8 · 2024-11-03

**Soundness:** 3
**Presentation:** 3
**Contribution:** 2
**Rating:** 5
**Confidence:** 4

**Summary:**

The paper presents an approach to initialize ViT through structured initialization of attention maps. By incorporating CNN-like inductive biases during initialization, it aims to combine the local spatial processing capabilities with the global relationship learning of attention mechanisms and take advantage of CNNs' inductive bias. Experimental results on several small-scale datasets validate its effectiveness.

**Strengths:**

- The paper is easy to follow.
- As much work has been trying to introduce convolutional design into the ViT model, this paper provides an interesting viewpoint that initializing the attention map as CNNs can also help to introduce the inductive bias and subsequentially improve the performance of trained ViT on small-scale datasets.
- A theoretical explanation is provided to show the connection between the structural initialization in ViT and inductive bias in CNNs.
- Some special designs like more heads and various initialization conv kernel sizes are adopted.

**Weaknesses:**

- The fundamental approach of forcing attention maps to mimic convolutional kernels seems to contradict the core advantage of attention mechanisms, as their advantage is to learn flexible, dynamic global relationships. It would be better to justify why structured initialization is preferred over simply incorporating convolutional blocks into the architecture, which would be a more straightforward solution.
- It would be better to provide more analysis of why this approach is better compared to well-established solutions:
  - Transfer learning from large-scale pre-trained ViTs
  - Hybrid architectures combining convolution and attention
- The optimization process required for initializing attention maps introduces additional computational overhead during training, and one needs to further choose the optimizer for initialization and conv kernel size (as well as other hyperparameters for different model sizes), which makes it impractical.

**Questions:**

-  Does the structured initialization limit the model's ability to learn better representation? It would be better to provide some representation-level analysis using metrics like Centered Kernel Alignment (CKA) similarity [1].

[1] Kornblith, Simon et al. “Similarity of Neural Network Representations Revisited.” ICML 2019.

---

> ### Author Response · Authors · 2024-11-19
> **Response to reviewer aWu8**
>
> We are encouraged to know that the reviewer thinks our paper is *“easy to follow”*, and that our method *“provides an interesting viewpoint”* on *“initializing the attention map as CNNs”* to *“introduce the inductive bias”*. We also thank the reviewer for acknowledging the contributions of the theoretical analysis and various ablation studies. We provide our main clarifications below.
>
>
> **Q: Does introducing CNN structural biases to Transformer contradict the advantage of attention mechanisms?**
>
> **A:** We would like to clarify that our method focuses on introducing the **architectural inductive bias** from CNNs to initialize the attention map **without changing the Transformer architectures**. We have emphasized this argument in the abstract and introduction, stating that, unlike previous arts that directly introduce convolutions into attention (*which is also referred to by the reviewer as a more straightforward solution*), potentially damaging the structural advantages of Transformers, our method only introduces architectural inductive bias in attention map initialization, **maintaining the inherent structural flexibility in ViTs**. Therefore, our method **preserves the Transformer architectures** and still allows the ViTs to learn flexible, dynamic global relationships. This is also one of the central innovations of our method. In addition, we have also provided experiments on large-scale applications to validate the stable performance of our method that preserves the architectural flexibility of ViTs.
>
>
> **Q: Why the proposed method is better than transfer learning on large-scale pre-trained models or hybrid architecture combining CNN and attention?**
>
> **A:** As stated in the introduction, we would like to emphasize the advantages of our proposed structured initialization method: 1. unlike previous methods that do transfer learning on large-scale pre-trained ViTs, our method involves **no pre-training of large-scale models on large-scale datasets** which may not be readily available; 2. our method **shares the advantages of both CNNs and ViTs**. To elaborate on this, we kindly refer the reviewer to the theoretical analysis in Section 3 on random filters and Appendix Section B on the convolutional representation matrix for theoretical explanations. In general, introducing inductive bias in CNNs for initializing the attention map helps ViTs start learning from a reasonable point where the original random initialization can lead the training to a noisy start especially when only small-scale datasets are available. Yet, our initialization method does not alter the structure of the ViTs, on the other hand, maintains the advantages of the Transformer architectures in learning dynamic, long-dependent global features. However, methods that directly combine the architectures of CNNs and ViTs will eventually alter the Transformer architectures, thus potentially damaging the architectural advantages of ViTs.
>
>
> **Q: Does pre-optimization of initialization introduce additional computation overhead?**
>
> **A:** We kindly refer the reviewer to Section 4.2 Line 289-291, where we stated that our pre-optimization is **not a pretrained step since no real data is involved**. Also, our optimization algorithm is very simple, and it serves as a surrogate for the SVD solver, converging in just a few seconds, *i.e.*, **around 5 seconds**. In addition, it is worth noting that for each embedding dimension and kernel size, the initialization only needs to be calculated once. As a simple optimization for the SVD solver, no careful choice of hyperparameters such as different optimizers or kernel size is needed.
>
>
> **Q: Does the structured initialization limit the model's ability to learn better representation?**
>
> **A:** As previously mentioned, since our method does not change the Transformer architectures, the structured initialization will not limit the models’ learning ability. We have shown similar or better performance of our methods compared to ViTs with default initializations tested on large-scale applications on ImageNet-1K.

---

> > ### Comment · Reviewer_aWu8 · 2024-11-27
> >
> > Thanks for the detailed feedback. After reading the rebuttal and comments from other reviewers, my question remains. The improvement on the widely-used ImageNet benchmark appears marginal. Moreover, as shown in Figure 6, the attention maps of the model initialized with structured initialization lean toward a local attention pattern, even after training. It's unclear whether this localized attention will not limit the models' learning ability, and the current analysis lacks quantitative validation beyond classification performance. The viewpoint is interesting, and I believe this work could be strengthened with more in-depth analysis. In its current version, I would categorize this paper as borderline.

---

> > > ### Author Response · Authors · 2024-11-27
> > > **Further clarifications for reviewer aWu8**
> > >
> > > We really appreciate the reviewer for providing additional discussions. We would like to clarify further based on the comments. We feel that the reviewer may misunderstand the focus of our method. We will address these concerns below.
> > >
> > > **Concern 1: the improvement on the widely-used ImageNet benchmark appears marginal**
> > >
> > > The reviewer is mentioning **Table 2** result where we have shown comparisons between our method and other initialization methods across different datasets. In short, it is reasonable that our method has a big improvement when training on small-scale data, and has on-par performance with little to no improvement when training on large-scale data. We have copied Table 2 below for better reference.
> > >
> > > | Method |  CIFAR-10 | CIFAR-100 | SVHN | ImageNet-1K |
> > > | :-:| :-: | :-: | :-: | :-: |
> > > | Kaiming Uniform (He et al., 2015) | 86.36 (*2.27* $\downarrow$) | 63.50 (*3.00* $\downarrow$) | 94.51 (*1.31* $\uparrow$) | 74.11 (*0.69* $\uparrow$) |
> > > | Trunc Normal | 88.63 | 66.50 | 93.20 | 73.42 |
> > > | Mimetic (Trockman & Kolter, 2023) | 91.16 (*2.53* $\uparrow$) | 70.40 (*3.90* $\uparrow$) | **97.53** (*4.33* $\uparrow$) | 74.34 (*0.92* $\uparrow$) |
> > > | Ours (Imp.-3) | **91.62** (*2.99* $\uparrow$) | 68.81 (*2.31* $\uparrow$) | 97.21 (*4.01* $\uparrow$) | 74.24 (*0.82* $\uparrow$) |
> > > | Ours (Imp.-5) | 90.67 (*2.04* $\uparrow$) | **70.46** (*3.96* $\uparrow$) | 97.23 (*4.03* $\uparrow$) | **74.40** (*0.98* $\uparrow$) |
> > >
> > > We would like to remind the reviewer that the focus of our method is not to improve the performance of the ViTs when trained on large-scale datasets such as ImageNet-1K. Instead, we have stated in our abstract (Line 011-017): **“The application of Vision Transformers (ViTs) to new domains where an inductive bias is known but only small datasets are available to train upon is a growing area of interest.”** We are aiming to solve the ViT training issues on small-scale datasets when large-scale data is not readily available. Therefore, we propose to introduce the inductive bias in the initialization in ViTs, and test our method on several small-scale data, proving **the big improvement of our method when training ViT on small-scale datasets**. However, we also provided results of using our initialization method and training on large-scale datasets such as ImageNet-1K to validate that although we introduce the CNN structure (locality) during initialization, our model **does not change the architecture of the ViTs** -- which is one of the central innovations of our method -- and the performance of our method trained on large-scale data is on par with the conventional initialized ViTs, and the performance did not degrade. In addition, when the dataset is large enough during training, ViTs could easily use the data to learn the features/patterns. Therefore, it is reasonable/foreseeable that introducing inductive bias of CNNs during initialization does not improve or degrade the performance of ViTs. Combining with the theoretical analysis, we believe we have validated our method **quantitatively and theoretically**.
> > >
> > > **Concern 2: the attention maps of the model initialized with structured initialization lean toward a local attention pattern, even after training**
> > >
> > > The reviewer mentions Figure 6 in the revised paper. In short, attention maps before and after training showing convolution pattern is reasonable and expected, it **does not indicate that our method cannot capture global dependency information**. We would also like to point out that the attention maps we showed in Figure 6 are analogous to figures shown in paper [c1] Figure 2 and paper [c2] Figure 7, where paper [c1] uses the mimetic initialization method, and paper [c2] uses pretrained weights from large ViTs trained on large-scale datasets. We would like to clarify that Figure 6 shows attention maps using our initialization method **both before training and after training**, and the attention maps are not normalized. In Figure 6, we can see that before training, since we introduce the inductive bias of CNNs into the attention map initialization of ViTs, our method offers off-diagonal attention peaks aligned with the impulse structures, validating the effect of our initialization. After training, our method still shows some of the convolution patterns, and it also shows that our method is able to capture global dependencies where almost all “patch patterns” in the attention map are attended. Combined with the on-par performance of our method on the large-scale dataset ImageNet-1K, it validates that our method is still able to perform well on large-scale data and captures global dependency information.
> > >
> > > **Reference:**
> > >
> > > [c1] Trockman, Asher, and J. Zico Kolter. "Mimetic initialization of self-attention layers." International Conference on Machine Learning. PMLR, 2023.
> > >
> > > [c2] Zhang, Yi, et al. "Unveiling transformers with lego: a synthetic reasoning task." arXiv preprint arXiv:2206.04301 (2022).
> > >
> > > We are happy to discuss further.

---

> ### Author Response · Authors · 2024-11-21
> **Do you have further questions?**
>
> Dear reviewer aWu8, we have provided the rebuttal below. Did our response resolve all your questions? Since the discussion period is approaching, if you have further questions, please don't hesitate to ask. We are happy to resolve your concerns.

---

### Official Review · Reviewer_P9Fk · 2024-11-04

**Soundness:** 3
**Presentation:** 3
**Contribution:** 2
**Rating:** 5
**Confidence:** 4

**Summary:**

This paper addresses the challenge of applying Vision Transformers (ViTs) to new domains with small datasets, where Convolutional Neural Networks (CNNs) typically excel due to their inherent architectural inductive bias. The authors propose a novel approach that reinterprets CNN's architectural bias as an initialization bias for ViTs, termed "structured initialization." Unlike traditional ViT initialization methods that rely on empirical results or attention weight distributions, this method is theoretically grounded and constructs structured attention maps. The paper demonstrates that this structured initialization enables ViTs to achieve performance comparable to CNNs on small-scale datasets while retaining the flexibility to perform well on larger-scale applications. The proposed method shows significant improvements over conventional ViT initialization across several small-scale benchmarks, including CIFAR-10, CIFAR-100, and SVHN, and maintains competitive performance on large-scale datasets like ImageNet-1K.

**Strengths:**

1.Theoretical Foundation: The structured initialization method is based on solid theoretical analysis rather than just empirical results, providing a strong rationale for its effectiveness.

2.Performance Improvements: The method consistently shows significant performance improvements over conventional ViT initialization methods in small-scale datasets, which is a notable achievement.

**Weaknesses:**

1. In terms of innovation, the Transformer architecture was initially designed to minimize inductive bias. The author's attempt to incorporate structural biases from CNNs into the Transformer seems to go against the original intent of the Transformer design, which could be seen as a step backward for the evolution of Transformer models.

2. The variety of experimental backbones is somewhat limited. It would be beneficial to conduct experiments with DeiT or Swin-Transformer to compare results. Furthermore, aside from classification tasks, it would be interesting to test the method on detection or segmentation tasks to further evaluate its versatility and effectiveness.

**Questions:**

Why not apply structured initialization to the value (V) component of self-attention? Additionally, how are the feed-forward network (FFN) layers, normalization layers, and projection layers initialized in the proposed method?

---

> ### Author Response · Authors · 2024-11-19
> **Response to reviewer P9Fk**
>
> We thank the reviewer for their acknowledgment of our paper being *“theoretically grounded”*, *“based on solid theoretical analysis”*, and our structured initialization *“shows significant performance improvements over conventional ViT initialization methods in small-scale datasets”*, *“while retaining the flexibility to perform well on larger-scale applications”*. We will clarify our main arguments here.
>
>
> **Q: Does introducing CNN structural biases to Transformer contradict its structure?**
>
> **A:** We would like to clarify that our method focuses on introducing the **architectural inductive bias** from CNNs to initialize the attention map **without changing the Transformer architectures**. We have emphasized this argument in the abstract and introduction, stating that, unlike previous arts that directly introduce convolutions into attention, potentially damaging the structural advantages of Transformers, our method only introduces architectural inductive bias in attention map initialization, **maintaining the inherent structural flexibility in ViTs**. Therefore, our method **preserves the Transformer architectures** and still allows the ViTs to learn flexible, dynamic global relationships. This is also one of the central innovations of our method. In addition, we have also provided experiments on large-scale applications to validate the stable performance of our method that preserves the architectural flexibility of ViTs.
>
>
> **Q: More backbone experiments with Transformers like DeiT or Swin-Transformer and more tasks other than classification?**
>
> **A:** We believe that the current experimental settings with **vanilla Transformer backbone** have already validated the effectiveness of our proposed methods on various small-scale datasets and large-scale applications through different analysis and ablations studies. We have also provided a solid theoretical analysis of our method. In addition, we would like to clarify that Transformer backbones such as Swin-Transformer have incorporated the structural inductive bias in CNN directly into the ViTs, contradicting the main contributions of our method that aims **not to modify the Transformer architectures, thus preserving the structural advantages of the ViTs**.
>
>
> **Q: Why not initialize value V? How are other components initialized?**
>
> **A:** We would like to remind the reviewer that our proposed initialization method theoretically originated from the architecture of CNNs, while in CNN structure, there is no reference to guide the initialization of value $V$. In addition, unlike previous methods that initialize $Q$, $K$, and $V$ separately, initializing the attention map itself ensures the incorporation of architectural inductive bias of CNNs.
> We use the default initialization adopted in the conventional Transformer to initialize other components such as FNN, normalization layers, and projection layers.

---

> > ### Comment · Reviewer_P9Fk · 2024-11-22
> >
> > Thank you for the rebuttal. After reading it, my major concerns remain unresolved, and thus I maintain my original score.

---

> > > ### Author Response · Authors · 2024-11-23
> > > **Response to reviewer P9Fk**
> > >
> > > We appreciate your feedback and understand that our previous response did not fully resolve your concerns. If you could be more specific about which concerns, we could provide more clarifications.

---

> ### Author Response · Authors · 2024-11-21
> **Do you have further questions?**
>
> Dear reviewer P9Fk, we have provided the rebuttal below. Did our response resolve all your questions? Since the discussion period is approaching, if you have further questions, please don't hesitate to ask. We are happy to resolve your concerns.

---

### Author Response · Authors · 2024-11-19
**A common response to all reviewers**

We thank all reviewers for the thoughtful comments and questions. We would like to make a common response to clarify the advantage and innovation of our method.

**Q: Does introducing CNN structural biases to Transformer contradict the advantage of its structure?**

**A:** We would like to clarify that our method focuses on introducing the **architectural inductive bias** from CNNs to initialize the attention map **without changing the Transformer architectures**. We have emphasized this argument in the abstract and introduction, stating that unlike previous arts [c1, c2, c3] that directly introduce convolutions into attention, potentially damaging the structural advantages of Transformers, our method only introduces architectural inductive bias in attention map initialization, **maintains the inherent structural flexibility in ViTs**. Therefore, our method **preserves the Transformer architectures** and still allows the ViTs to learn flexible, dynamic global relationships. This is also one of the central innovations of our method. In addition, we have also provided experiments on large-scale applications to validate the stable performance of our method that preserves the architectural flexibility of ViTs.

References (also cited in our paper):

[c1] Yuan, Kun, et al. "Incorporating convolution designs into visual transformers." Proceedings of the IEEE/CVF international conference on computer vision. 2021.

[c2] Li, Kunchang, et al. "Uniformer: Unifying convolution and self-attention for visual recognition." IEEE Transactions on Pattern Analysis and Machine Intelligence 45.10 (2023): 12581-12600.

[c3] Dai, Zihang, et al. "Coatnet: Marrying convolution and attention for all data sizes." Advances in neural information processing systems 34 (2021): 3965-3977.


**Q: Does pre-optimization add computation overhead?**

**A:** We kindly refer the reviewer to Section 4.2 Line 289-291, where we stated that our pre-optimization is **not a pretrained step since no real data is involved**. Also, our optimization algorithm is very simple, and it serves as a surrogate for the SVD solver, converging in just a few seconds, *i.e.*, **around 5 seconds**. In addition, it is worth noting that for each embedding dimension and kernel size, the initialization only needs to be calculated once. As a simple optimization for the SVD solver, the optimization will converge, and no careful choice of hyperparameters such as different optimizers or kernel size is needed.


We have also provided individual responses below. If you have further questions, please give us comments and we are happy to discuss.

---

### Author Response · Authors · 2024-11-24
**A revised version of the paper is submitted**

We have submitted a revised version with all changes marked in red. Specifically, we have emphasized our arguments in the introduction and contribution sections. Additional figures of attention maps before and after training are added to Appendix Figure 6 for further clarification.

---

### Author Response · Authors · 2024-12-02
**An implementation of our method shared with all reviewers**

Dear reviewers, we have shared an anonymous Colab link with an implementation of our method here: [https://colab.research.google.com/drive/1Oic07RhIbqKUcVoCQnd2A2c73LBtdSq6?usp=sharing](https://colab.research.google.com/drive/1Oic07RhIbqKUcVoCQnd2A2c73LBtdSq6?usp=sharing).

In this notebook, we have provided all training details for the three main initialization methods that are compared in our paper: **random** ([timm default](https://github.com/huggingface/pytorch-image-models)) initialization, **mimetic** initialization ([Trockman & Kolte](https://proceedings.mlr.press/v202/trockman23a/trockman23a.pdf)), and our **structured initialization** ([ours](https://openreview.net/pdf?id=z9UBpl4pv5)) method.

We only trained for **20 epochs** to save time. Note that in our paper (real training), we train for **200 epochs** with **all the other settings (hyperparameters) *the same as* the settings defined in this notebook**.

We have also included a summary of (1) initialization (pre-optimization) time, (2) pre-training attention maps, and (3) post-training attention maps. We hope it will further clarify the concerns that reviewers have raised.

Please also note that the initialization time is directly computed using the **Colab T4 GPU**, therefore, it is different from what we have reported in our paper (**$\sim$5 seconds** for our initialization method). Nonetheless, the pre-optimization time (in seconds) is neglectable compared to the long training time which can last for hours or even a few days.

Please let us know if you have further questions or concerns.

---

### Meta-Review · Area_Chair_oRnS · 2024-12-17

**Metareview:**

This paper proposes structured initialization to improve the performance of Vision Transformers (ViTs) on small-scale datasets. The approach reinterprets architectural bias from CNN as an initialization bias for ViTs. This method enables ViTs to achieve competitive performance with CNNs on small datasets, such as CIFAR-10, CIFAR-100, and SVHN, while maintaining flexibility and strong performance on larger datasets like ImageNet-1K. The paper provides both theoretical justification and empirical validation, demonstrating improvements over conventional ViT initialization methods.

After the review process, all reviewers unanimously voted for rejection, citing several concerns, including limited experiments, marginal improvements on large datasets, and the generalizability of the approach to other datasets.

**Additional Comments On Reviewer Discussion:**

All reviewers responded to the authors' rebuttal and leaned toward rejection.

---

### Decision · Program_Chairs · 2025-01-22

Reject